# Computational Fluid Dynamics Modeling and Field Applications of Non-Powered Hydraulic Mixing in Water Treatment Plants

**Tea In Ohm, Jong Seong Cae, Meng Yu Zhang and Jin Chul Joo \*** 

Department Civil and Environmental Engineering, Hanbat National University, Daejeon 34158, Korea; tiohm1@hanbat.ac.kr (T.I.O.); airpc@hanmail.net (J.S.C.); zhangmengyu11@gmail.com (M.Y.Z.)
\* Correspondence: jincjoo@hanbat.ac.kr; Tel.: +82-42-821-1264

**Abstract:** In this study, non-powered hydraulic mixing with three layers of baffles and holes was evaluated as an alternative to vertical shaft impellers in a rapid mixing process through both computational fluid dynamics (CFD) modeling and field applications. From the CFD modeling, the turbulence (i.e., vortex rings) caused by excess kinetic energy between the inlet and second-layer baffle ensures rapid mixing of the coagulants throughout the total water flow and overcomes the damping effect of the components in a mixing basin. Although optimal inlet velocity needs to be investigated for sufficient mixing between coagulants and pollutants in raw water with relatively low energy consumption and maintenance costs, non-powered hydraulic mixing developed in this study was proved to create strong turbulence and can be applied in any water treatment plants that involves coagulation-flocculation processes. Based on the comparison of the water quality between two water treatment plants using identical raw water and coagulant operated from 2014 to 2016, no difference in water quality of treated water indicated that non-powered hydraulic mixing can be replaced with vertical shaft impellers, hence, both energy consumption and maintenance costs can be reduced. Further study is warranted to optimize non-powered hydraulic mixing for the tradeoff between mixing efficiency and energy consumption in the water treatment plants.

**Keywords:** coagulation-flocculation processes; computational fluid dynamics (CFD), mixing efficiency; non-powered hydraulic mixing; water treatment plant; turbulence

## 1. Introduction

Water treatment processes and plants vary with both type and quality of raw water, and directly affect the overall efficiency of the water treatment systems. Generally, unit operations and unit processes commonly used in water treatment systems include screening, pumping, aeration, chemical injection, rapid mixing, coagulation-flocculation, filtration, and disinfection [1]. Among various unit processes, the mixing process in water treatment systems is a vital step in which certain coagulant is rapidly dispersed through raw water and reacted with various pollutants to form coagulated and precipitable flocs [2–4]. Both coagulation by rapid mixing and flocculation by slow mixing are integral treatment steps in which pollutants undergo the transition from steady to unsteady states (e.g., destabilization of colloids, removal of organic and inorganic matter, removal of metals and anions, and removal of pathogen microorganisms) [1–5].

The efficiency of both coagulation and flocculation processes has been reported to be affected by various factors such as mixing intensity and time, coagulant dosing, temperature, pH, and transit time [2–13]. Thus, numerous studies have investigated the complex interactions of each factor affecting the efficiency of both coagulation and flocculation processes [2–13]. Among the various factors, rapid

mixing is the first step of water treatment systems and is required to achieve instantaneous and uniform dispersion of coagulants throughout the whole water body.

In the rapid mixing processes (e.g., mechanical mixer, in-line static mixer, in-line mechanical blender, jet injection blending, and hydraulic jumps), certain coagulant is injected to suspended particles ranged from 0.001 to 1 μm and mixed rapidly to form flocs that can be deposited and filtered [1,5,7]. Although both type and dosage of coagulants have significant impacts on water treatment efficiency [4,6,7,11,12], the efficiency of rapid mixing process can be enhanced by optimization of the velocity gradient (*G*) and the contact time (*t*). Accordingly, both rapid mixing (*G*) between raw water and coagulant to increase the dispersion rate and adequate contact time (*t*) to promote the particle collisions are critical factors to determine the efficiency of coagulation-flocculation processes in water treatment systems.

Most water treatment plants in Republic of Korea have implemented mechanical mixers to mix coagulants (i.e., aluminium sulfate, polyaluminium chloride, polyaluminium chlorosulphate, etc.) using vertical shaft impellers, which do not allow rapid mixing of certain coagulants due to significant head loss and short-circuiting. These vertical shaft impellers are disadvantageous in terms of maintenance with high energy costs and high risk of mechanical failures. Also, the rotational speed of vertical shaft impellers can cause the breakup of the flocs, and the power system induces vibration, noise and excessive operation and maintenance costs. Thus, the chance of colliding with pollutants needs be increased by injecting the excessive amount of coagulants by 30–40% [5,11]. Consequently, conventional mechanical rapid mixing using vertical shaft impellers needs to be improved to increase the mixing efficiency and to reduce both energy consumption and maintenance costs.

To provide the rapid dispersion of certain coagulants throughout the raw water with minimal head loss and low cost, widely-implemented alternatives to vertical shaft impellers are the application of hydraulic mixing with baffles or throttling valves and pump blenders mixing through a diffuser in a pipe [1,14–18]. In this study, non-powered hydraulic mixing with three-layers of baffles and holes was designed to optimize the turbulence (i.e., vortex rings) in raw water as a rapid mixing method to complete the reaction kinetics of coagulants with the pollutants.

Considering that hydraulic mixing may not cause enough turbulence ensuring rapid dispersion of the coagulants throughout the total water flow if the flow rate through the system is reduced, the operational flexibility of the non-powered hydraulic mixing needs to be investigated. Thus, computational fluid dynamics (CFD) was applied to estimate the head loss, velocity distribution and turbulence intensity as a function of the flow rate of raw water, and to optimize the design of the non-powered hydraulic mixing with high flexibility in operation [19–21].

The specific objectives of this study were to investigate the effect of inlet velocity (i.e., flow rate of raw water) on the velocity distribution, the internal pressure, and the turbulence intensity of non-powered hydraulic mixing by using CFD estimation, and to analyze the water treatment efficiency, the energy saving, and the feasibility of non-powered hydraulic mixing by comparison of two water treatment plants with different types of rapid mixing processes (plant A: non-powered hydraulic mixing vs. plant B: mechanical shaft impellers) using identical raw water and coagulant (i.e., aluminum sulfate (VI) [$Al_2(SO_4)_3 \cdot 18H_2O$]).

## 2. Methods and Modeling

### 2.1. Non-Powered Hydraulic Mixing Design

With the aim of replacing the existing mechanical rapid mixers in water treatment plants with non-powered hydraulic mixing, CFD was utilized to obtain the design factors (size, depth, hole size, and wing shape of the baffles) [19–21]. The non-powered hydraulic mixing was designed to optimize the turbulence (i.e., vortex rings) in flowing water and did not require separate power systems. Instead, both baffles and holes are installed vertically along the flow direction of flowing water, thereby creating strong turbulence to mix both coagulants and pollutants in water. The optimized

non-powered hydraulic mixing mainly consists of an inlet, outlet, and internal circulation components with three-layer baffles and many holes.

To prevent the incomplete mixing between coagulants and pollutants in raw water, each wing of the baffles and the frames was inclined at an angle of 30° to the horizontal plane, and many holes were drilled to minimize the resistance to raw water, as illustrated in Figure 1. The raw water inlet at the lower part of the mixer is a cylinder with a diameter of 800 mm, and the mixing water outlet, which is located at the upper part of the mixer, has a rectangular area of 1107 × 1107 mm.

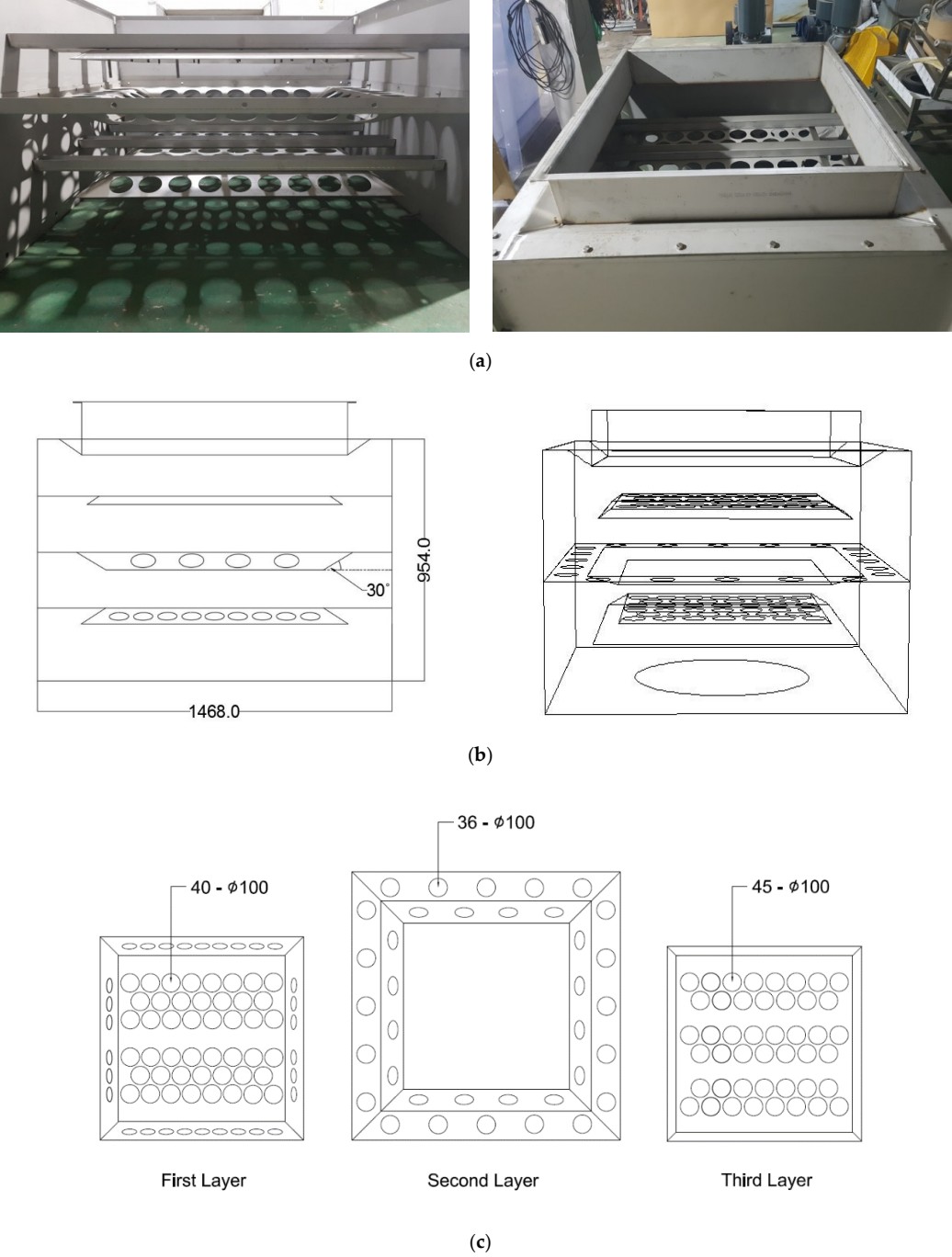

**Figure 1.** Pictorial view and schematic diagram of non-powered hydraulic mixing with three layers of baffles and holes. (**a**) Pictorial view; (**b**) horizontal sectional view; (**c**) plan view of each layer.

The baffles of the first and third layers within the internal circulation components were fabricated with lower edges, hence, there is space for a flow path between each baffle and the inner wall of the mixing basin. The first-layer baffle generates 100-mm vortex rings around 40 holes, and the wing of the first-layer baffle inclined at an angle of 30° to the horizontal creates vortex rings with various sizes near the inner wall of the mixing basin. The cavity at the center of the second-layer baffle ensures that raw water collides with coagulants and forms into strong vortex rings with various sizes, increasing the chance of colliding between coagulants and pollutants. After the mixing process through the second layer, adequate time was provided to allow the formation of micro-size flocs in the third-layer baffle before discharge. To minimize the significant pressure and head loss due to the components inside the mixing basin, the outlet area was designed to be 130% larger than the inlet area.

### 2.2. Computational Fluid Dynamics (CFD) Modeling Method

#### 2.2.1. Governing Equation

This study utilized the Navier–Stokes equation as the governing Equation (1) for 3D steady-state flow using ANSYS-FLUENT Ver.16.1. [22].

$$\frac{\partial(\rho_i \Phi_i)}{\partial t} + \nabla \cdot (\rho_i \Phi_i U_i) = -\nabla \cdot (\Gamma \nabla \Phi_i) + S_\Phi \tag{1}$$

where $\rho_i$ refers to density (kg/m$^3$), $t$ is time (sec), $\Phi_i$ is dependent variable (velocity, enthalpy, species mass fraction), $U_i$ is a velocity component (m/sec), and $S_\Phi$ is the source term. The most widely used $\kappa - \varepsilon$ model was applied to analyze turbulence. The turbulence shear stress was expressed in terms of the product of the turbulence viscosity ($\mu_t$) and mean velocity by the Boussinesq assumption. The turbulence viscosity was calculated from the Prandtl–Kolmogorov relationship, as displayed in Equation (2) [23,24]:

$$\mu_t = \frac{\rho C_\mu \kappa^2}{\varepsilon} \tag{2}$$

where $C_\mu$ is empirical constant. The turbulence eddy viscosity can be obtained from the governing equation of turbulence energy ($\kappa$) and the dissipation rate ($\varepsilon$) [23,24].

#### 2.2.2. Modeling and Mesh Creation

A Cartesian coordinate system was constructed with the origin at the center of the raw water inlet and $x$, $y$, and $z$ axes. Both raw water and coagulants were injected in the direction of the $z$ axis, and, vortex rings were generated in turbulence at both first- and second-layer baffles, enabling the rapid mixing between coagulants and pollutants. Then, at the third-layer baffle, the slow mixing allows the production of micro-size flocs in sedimentation process. A separate mesh was created for each step after validation of mesh systems in more complex calculations, and the boundary was adopted using the inlet velocity of 0.2–0.7 m/s at water temperature of 15 °C.

Figure 2 displays the optimized geometric shape of the non-powered hydraulic mixing after preliminary calculations for both coarse and dense meshes using GAMBIT 2.5.6. A tetrahedral/hybrid grid system was used for the mesh shape, and the total number of grids was 971,050. The grids at the inlet and around the holes of the baffles were set more densely to increase the analytical accuracy. The inlet velocity of raw water was set to 0.2~0.7 m/s to evaluate the effect of flow rate, and a semi-implicit method for pressure linked equations-consistent (SIMPLEC) algorithm was implemented to correct the pressure and velocity terms. In addition, the mixer inlet is perpendicular to the flow velocity, and the no-slip condition was applied to the inner walls, the surfaces of the baffles, and the frames in the mixer.

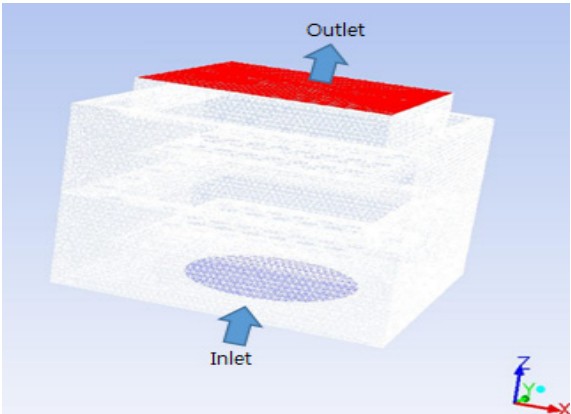

**Figure 2.** Mesh structure of non-powered hydraulic mixing with three layers of baffles.

For the tetrahedral/hybrid grid system developed in this study, a steady-state solution of the governing equation was obtained iteratively using the solver to couple velocity and pressure, and iterations were terminated when the maximum residual of the momentum equations and $\kappa - \varepsilon$ equations were less than $10^{-4}$. The convergence criterion of the velocities ($u$, $v$, $w$), kinetic energy ($\kappa$) and kinetic energy dissipation ($\varepsilon$) is $10^{-3}$, respectively, and that of continuity is $10^{-2}$. From this study, the optimized values of the under-relaxation factors were in the range of 0.3–0.7 at the end of 5500 iterations.

### 2.2.3. Calculation of Velocity Gradient ($G$) and Head-Loss Rate

In the mixing process, the coagulant must be introduced at points of high turbulence in raw water to complete both coagulation and flocculation processes with pollutants within several seconds. Thus, the mixing efficiency can be determined mainly by the efficiency of rapid mixing based on the power imparted to the water. In this study, turbulence intensity ($I$) is defined as the ratio of the root-mean-square of the velocity fluctuations ($u'$) to the mean flow velocity ($u_{avg}$), as displayed in Equations (3)–(5):

$$I = \frac{u'}{u_{avg}} \tag{3}$$

$$u' = \sqrt{\frac{1}{3}\left(u'^2_x + u'^2_y + u'^2_z\right)} \tag{4}$$

$$u_{avg} = \sqrt{\left(u^2_x + u^2_y + u^2_z\right)} \tag{5}$$

Since velocity gradient $G$ (s$^{-1}$) is used as the criterion for the mixing intensity of the rapid mixing, $G$ values can be calculated based on the input energy, the viscosity of fluid inside the mixing basin, and the volume of the mixing basin. The equation for calculating the $G$ value is given by Equation (6):

$$G = \sqrt{\frac{P}{\mu V}} \tag{6}$$

where $P$ is the mixing power (Watt) applied to water, $\mu$ is the viscosity (kg/m·s) of water, and $V$ is the volume (m$^3$) of the mixing basin.

When the water level of the grit chamber in the mixing basin increased to 30 cm, the head loss reached a value of 3000 Pa. Thus, the pressure loss between the inlet and outlet was calculated using Equation (7):

$$Head \quad Loss \quad Rate(\%) = \frac{pressure \quad loss(calculation)}{3000} \times 100(\%) \tag{7}$$

### 2.3. Water Treatment Plants with Different Mixing Method

Both water treatment plants A and B used identical raw water from Lake D located at high elevations in mountain ranges. Lake D is generally mesotrophic with a trophic state index of 40–50 and contains moderate amounts of organics and nutrients with occasional algal blooms during the end of summer. Details of water quality of raw water supplied to both water treatment plants with different mixing methods during the experiment are summarized in Table 1. Under this condition, various water-quality parameters were analyzed, and compared between plants A and B to investigate the feasibility of the non-powered hydraulic mixing in water treatment plants.

**Table 1.** Water quality of raw water supplied to both water treatment plants.

| Parameters | pH | BOD [a] (mg/L) | TOC [b] (mg/L) | SS [c] (mg/L) | DO [d] (mg/L) | Chl-*a* [e] (mg/m$^3$) | T-P [f] (mg/L) | T-N [g] (mg/L) |
|---|---|---|---|---|---|---|---|---|
| Values | 7.9 | 0.91 | 2.40 | 1.90 | 10.4 | 10.6 | 0.012 | 1.19 |

[a] Biochemical Oxygen Demand; [b] Total Organic Carbon; [c] Suspended Solids; [d] Dissolved Oxygen; [e] Chlorophyll-*a*; [f] Total-Phosphorus; [g] Total-Nitrogen.

Whereas Plant A implemented the developed non-powered hydraulic mixing with plant capacity of 600,000 m$^3$/day as displayed in Figure 3a, Plant B used the conventional mechanical vertical shaft impellers with plant capacity of 300,000 m$^3$/day as displayed in Figure 3b. Both water treatment plants used conventional treatment processes such as coagulation, flocculation, clarification, and filtration, and disinfection at full scale, and the optimal pH for coagulation was near 6.5. For Plant A with inlet velocity of 0.38 m/s, a total of 12 vertical shaft impellers were replaced by non-powered hydraulic mixing and operated from 2014 to 2016. Typical velocity gradients generated in the conventional mechanical vertical shaft impellers were 500–600 s$^{-1}$ with residence times of 60 s whereas estimated velocity gradients generated in non-powered hydraulic mixing were 600–800 s$^{-1}$ with residence times of 30 s.

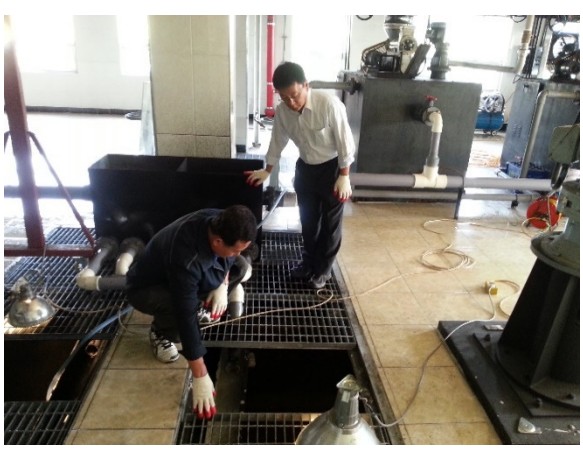
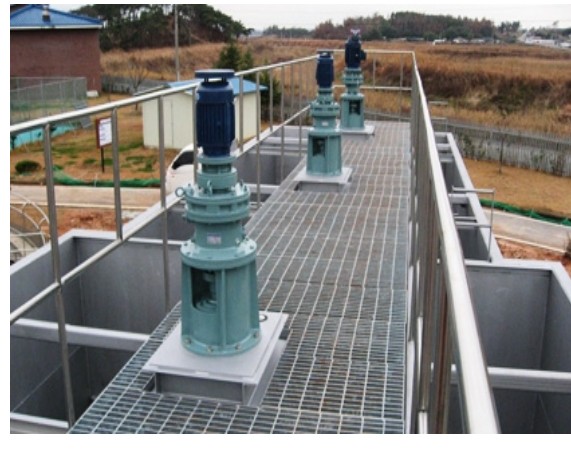

(**a**)																																				(**b**)

**Figure 3.** Pictorial view of water treatment plants with different rapid mixing methods. (**a**) Non-powered hydraulic mixing; (**b**) mechanical vertical shaft impellers.

Both plants have used same dosages (i.e., 50 mL/m$^3$) of commercial aluminum sulfate (VI) [Al$_2$(SO$_4$)$_3$·18H$_2$O; 8.3% *w/w* for liquid Al$_2$O$_3$; Unicon chemica] Co., India] for chemical coagulation in the presence of sufficient alkalinity (i.e., calcium bicarbonate) under the control of relatively similar pH values near 6.5 and water temperature around 15 °C. Also, both plants have implemented identical rapid sand filtration process with higher filtration rate of 10 m$^3$/m$^2$/h and disinfection process with direct injection of liquid chlorine equivalent to 1 mg-Cl$_2$/L in water.

## 3. Results and Discussion

### 3.1. Effect of Inlet Velocity on Velocity Distribution

Figure 4 displays the velocity distribution changes as a function of the inlet velocities ranging from 0.2 to 0.5 m/s. As the inlet velocity of raw water increased, the water velocity increased at the holes of the baffles, the walls of the mixing basin, and the inlet. In addition, wider velocity distribution and stronger turbulence were observed in the lower part of the mixing basin, where the rapid mixing between coagulants and pollutants in raw water occurred. Considering the wider velocity distribution and stronger turbulence generated by non-powered hydraulic mixing, non-powered hydraulic mixing can replace the mechanical rapid mixers.

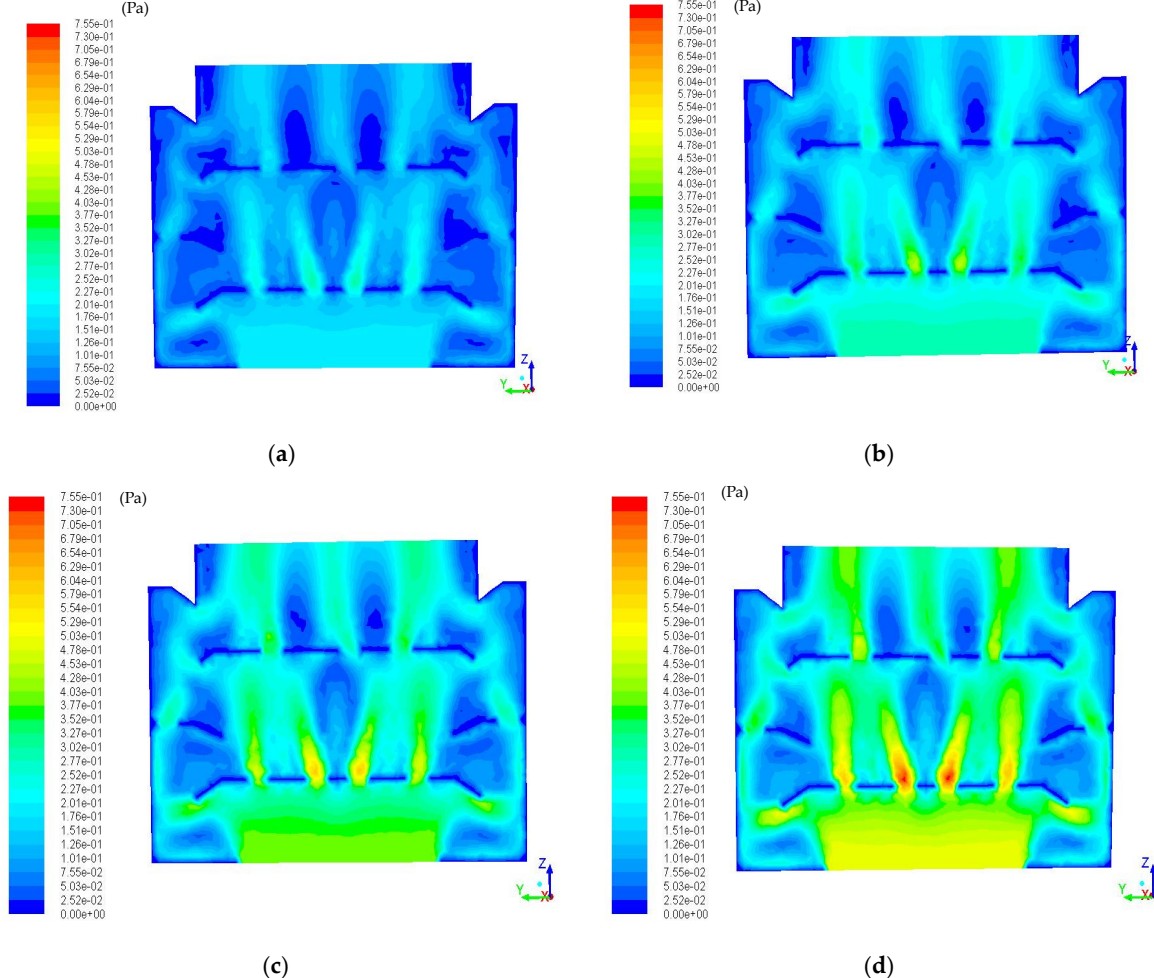

(a)　　　　　　　　　　　　　　　　　　　　　　(b)

(c)　　　　　　　　　　　　　　　　　　　　　　(d)

**Figure 4.** Computational fluid dynamics (CFD)-simulated velocity distribution changes in non-powered hydraulic mixing with the different inlet velocities. (**a**) Inlet velocity: 0.2 m/s; (**b**) inlet velocity: 0.3 m/s; (**c**) inlet velocity: 0.4 m/s; (**d**) inlet velocity: 0.5 m/s.

Similar to this study, Haarhoff [25] reported that the variation of velocity gradient is significant in the around-the-bend hydraulic flocculators, and the *G* value closely related with the turbulent dissipation rate of kinetic energy. Also, Haarhoff and van der Walt [26] estimated the spatial variations of the velocity gradient and high turbulence at the edges of the baffles in the hydraulic flocculator using CFD modeling. Then, Haarhoff and van der Walt [26] insisted that the slot width ratio is the most important, followed by the overlap ratio and by the depth ratio.

Table 2 summarizes both changes and differences in water velocity at each layer of baffles with the different inlet velocities of raw water. For each inlet velocity of raw water, water velocities flowing

through the outlet were lower than those through the inlet. However, the water velocities through first-layer baffle were similar to those of outlet, indicating the turbulence caused by excess kinetic energy between the inlet and second-layer baffles ensures rapid mixing of the coagulants throughout the total water flow, and overcomes the damping effect of the components in mixing basin after second-layer baffle.

**Table 2.** CFD-simulated velocity changes for three layers of baffles and outlet inside the hydraulic mixing basin with different inlet velocities.

| Inlet (m/s) | 1st Layer (m/s) | 2nd Layer (m/s) | 3rd Layer (m/s) | Outlet (m/s) | Ratio of Outlet to Inlet Velocity |
|---|---|---|---|---|---|
| 0.2 | 0.11 | 0.07 | 0.08 | 0.09 | 0.45 |
| 0.3 | 0.15 | 0.10 | 0.11 | 0.14 | 0.47 |
| 0.4 | 0.20 | 0.14 | 0.15 | 0.18 | 0.45 |
| 0.5 | 0.25 | 0.17 | 0.19 | 0.23 | 0.46 |
| 0.6 | 0.31 | 0.21 | 0.23 | 0.29 | 0.48 |
| 0.7 | 0.39 | 0.26 | 0.28 | 0.32 | 0.53 |

As the inlet velocity of raw water increased, the difference in water velocity between inlet and outlet increased and the ratio of outlet to inlet velocity increased slightly. As a result, strong vortex rings in turbulence were generated from greater inlet velocity, and both pressure and head loss became greater as the inlet velocity of raw water increased. Thus, optimal inlet velocity needs to be investigated for enough mixing between coagulants and pollutants in raw water with relatively low energy consumption and maintenance costs.

*3.2. Effect of Inlet Velocity on the Internal Pressure*

Considering that the water treatment plants implementing the non-powered hydraulic mixing can change the internal pressure and the head in water as a function of the inlet velocity of raw water, it is necessary to calculate the corresponding pressure and head losses at the inlet and outlet. The variations in head loss caused by the changes in inlet velocities are compared in Table 3, and the pressure distribution changes with different inlet velocities ranging from 0.2 to 0.5 m/s are displayed in Figure 5.

**Table 3.** CFD-simulated changes in both pressure and head with different inlet velocities.

| Inlet (m/s) | Inlet Pressure (Pa) | Outlet Pressure (Pa) | Pressure Loss (Pa) | Head Loss Rate (%) |
|---|---|---|---|---|
| 0.2 | 58.15 | 5.56 | 52.59 | 1.75 |
| 0.3 | 134.25 | 12.02 | 122.23 | 4.07 |
| 0.4 | 238.46 | 21.40 | 217.07 | 7.24 |
| 0.5 | 372.48 | 33.45 | 339.03 | 11.30 |
| 0.6 | 536.35 | 48.16 | 488.19 | 16.27 |
| 0.7 | 711.55 | 68.44 | 643.11 | 21.44 |

As summarized in Table 3, when the inlet velocity was 0.2 m/s, the inlet and outlet pressures resulted in 58.15 Pa and 5.56 Pa, respectively, thereby indicating that both pressure and head loss were 52.59 Pa and 1.75%, respectively. However, when the inlet velocity was 0.7 m/s, the inlet and outlet pressures resulted in 711.55 Pa and 68.44 Pa, respectively, thereby indicating that both pressure and head loss were 643.11 Pa and 21.44%, respectively.

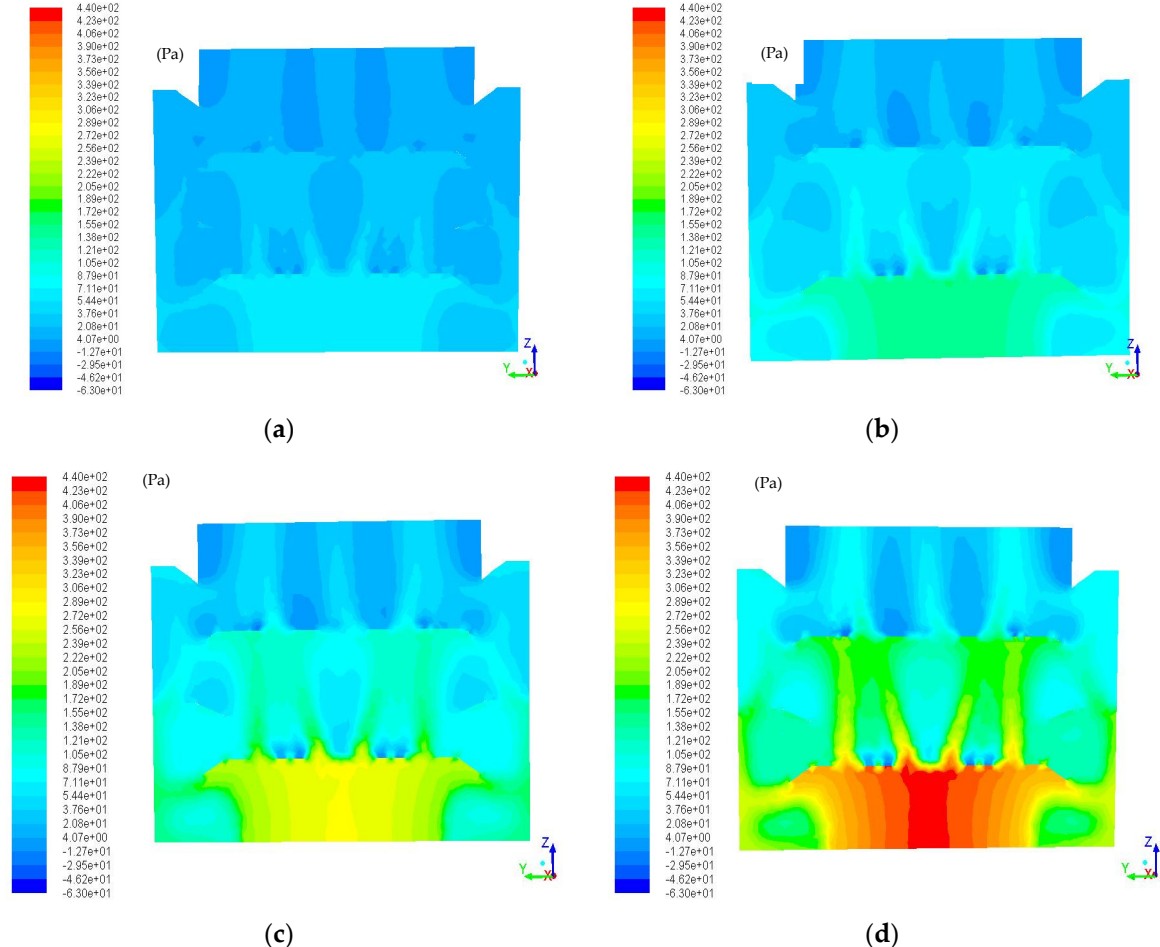

**Figure 5.** CFD-simulated pressure distribution changes in non-powered hydraulic mixing with different inlet velocities. (**a**) Inlet velocity: 0.2 m/s; (**b**) inlet velocity: 0.3 m/s; (**c**) inlet velocity: 0.4 m/s; (**d**) inlet velocity: 0.5 m/s

Consistent with the results from velocity distribution, as the inlet velocity increased, the loss in both pressure and head between the inlet and outlet increased, although the outlet area of the mixer was set to be 130% greater than the inlet area to minimize the loss in both pressure and head caused by the non-powered hydraulic mixing. These results indicated that the optimal inlet velocity needs to be determined for enough mixing between coagulants and pollutants in raw water without increase in both energy consumption and maintenance costs. Based on the appropriate design and analysis optimized by CFD modeling and the tradeoff between mixing efficiency and energy consumption, the optimal inlet velocity of 0.5 m/s was determined in this study to implement non-powered hydraulic mixing in the water treatment plants.

## 3.3. Effect of Inlet Velocity on the Turbulence Intensity

In the case of the non-powered hydraulic mixing, the *G* value cannot be obtained via the same equation used for the conventional mechanical mixer because the power value (*P*) of the non-powered hydraulic mixing is zero [27]. Thus, the mixing efficiency of the non-powered hydraulic mixing cannot be calculated based on mixing intensity. For this reason, in this study, the turbulence intensity was calculated according to the increase in the inlet velocity of raw water to evaluate the mixing intensity (see Table 4).

**Table 4.** CFD-simulated changes in turbulence intensity with different inlet velocities.

| Inlet Velocity (m/s) | Inlet Turbulence Intensity (%) | Outlet Turbulence Intensity (%) |
|:---:|:---:|:---:|
| 0.2 | 0.98 | 3.03 |
| 0.3 | 1.45 | 4.74 |
| 0.4 | 1.91 | 6.28 |
| 0.5 | 2.37 | 7.83 |

Considering that the magnitude of turbulence intensity was classified as high (5%~20%), medium (1%~5%), or low (less than 1%), Figures 6 and 7 illustrate the changes in turbulence intensity with different inlet velocities and the turbulence intensity pathlines at inlet velocity of 0.5 m/s, respectively. Whereas the inlet and outlet turbulence intensities for an inlet velocity of 0.2 m/s were 0.98% and 3.03%, respectively, the inlet and outlet turbulence intensities for an inlet velocity of 0.5 m/s were 2.37% and 7.83%, respectively.

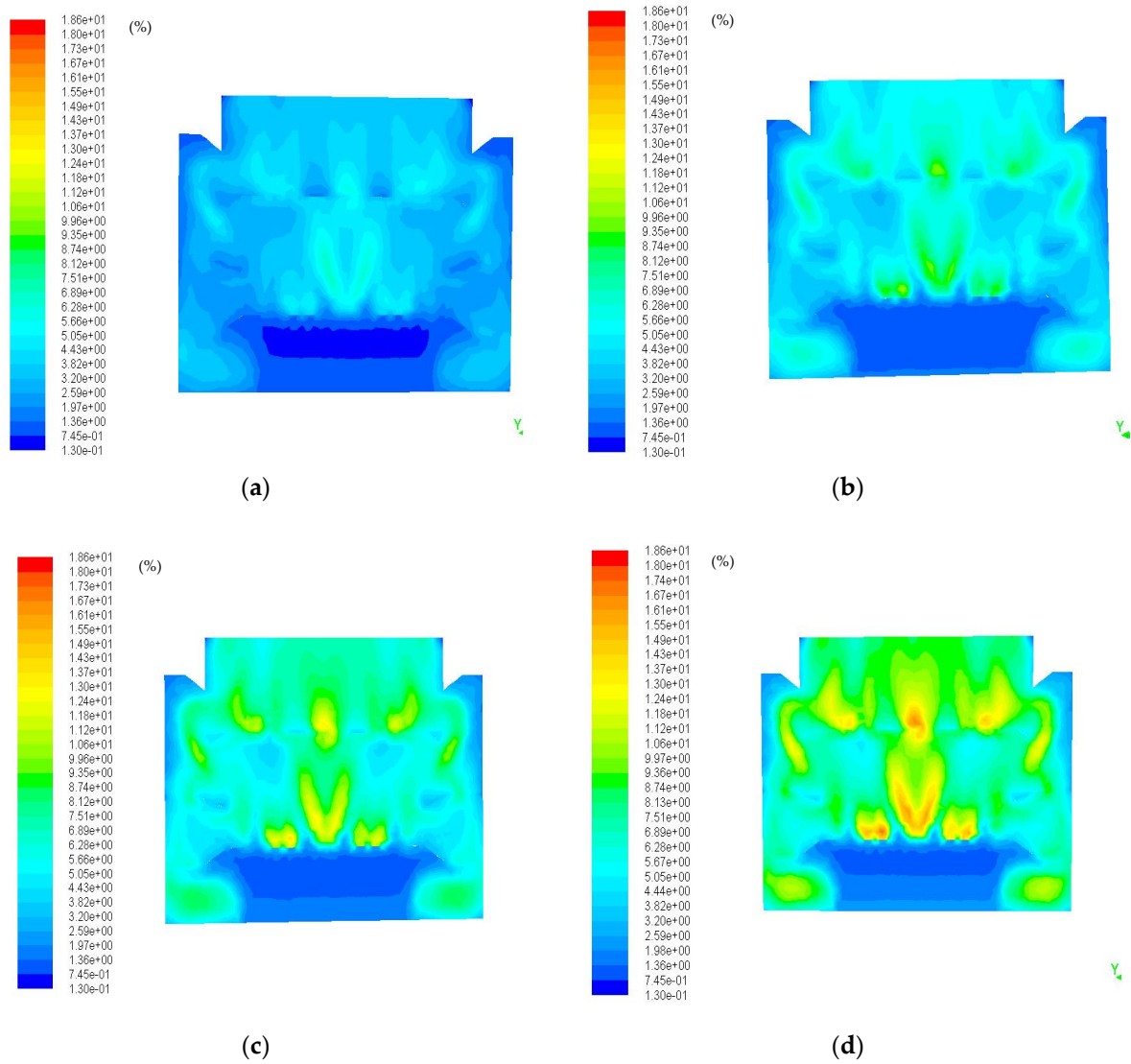

**Figure 6.** CFD-simulated turbulence intensity (%) changes in non-powered hydraulic mixing with different inlet velocities. (**a**) Inlet velocity: 0.2 m/s; (**b**) inlet velocity: 0.3 m/s; (**c**) inlet velocity: 0.4 m/s; (**d**) inlet velocity: 0.5 m/s.

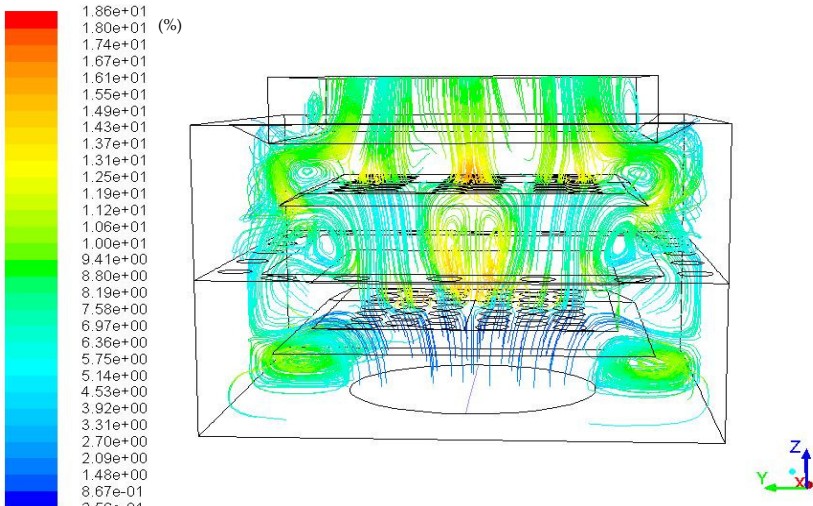

**Figure 7.** CFD-simulated turbulence intensity pathlines in non-powered hydraulic mixing at inlet water velocity of 0.5 m/s.

Consequently, stronger turbulence was estimated to occur between the first and second layers of baffles as the inlet velocity increased, indicating that the higher mixing intensities resulted from stronger turbulence. Also, non-powered hydraulic mixing using three layers of baffles and holes on baffles has enough water velocity to cause turbulence, ensuring the flash mixing of the coagulants throughout the total water flow.

Consistent with other studies [25,26], CFD optimized in this study can successfully estimate the flow behavior, the variation in local energy dissipation, and the velocity contours of hydraulic mixing in coagulation-flocculation processes of full-scale water treatment plants. Thus, CFD can be used to optimize systematically both velocity and pressure distribution and turbulence intensity and pathlines of hydraulic mixing in coagulation-flocculation processes using several critical design factors.

### 3.4. Results and Implications of Field Application in Water Treatment Plants

Table 5 presents a comparison of the water quality analysis of treated water between plant A with non-powered hydraulic mixing and plant B with vertical shaft impellers from 2014 to 2016. Some of the water quality parameters such as ammonia nitrogen, total phosphorous, total coliform, and fluorine were not detected for three years; moreover, no bacteria were detected. As summarized in Table 5, no significant difference in water quality between two water treatment plants using identical raw water and coagulant indicated that non-powered hydraulic mixing can be replaced with vertical shaft impellers, hence, both energy consumption and maintenance costs can be reduced.

Furthermore, all water-quality parameters of both plants were observed to be less than water quality standards, noting that no change in water quality was observed after replacing the mechanical vertical shaft impellers with non-powered hydraulic mixing. Considering that the velocity gradient (*G* value) of the mechanical vertical shaft impellers, which had been used in Plant A before being replaced by a non-powered vortexes mixer without energy consumption, was about 522 s$^{-1}$ [28], the 64,143~65,306 kWh/year (175.7~178.9 kWh/day) of energy consumed by each vertical shaft impeller (capacity: 50,000 m$^3$/day) could be saved. Finally, since the reduction of energy and maintenance costs in both CFD analysis and field applications without affecting the overall water quality were validated, the non-powered hydraulic mixing using horizontal jumps and flow energy developed in this study can replace the vertical shaft impellers in water treatment plants.

**Table 5.** Comparison of average water quality of treated water from Plants A and B.

| Water Quality Parameters | Standard | 2014 | | 2015 | | 2016 | |
|---|---|---|---|---|---|---|---|
| | | A | B | A | B | A | B |
| Turbidity | Less than 0.5 NTU [a] | 0.05 | 0.05 | 0.07 | 0.08 | 0.04 | 0.05 |
| pH | 5.8–8.5 | 7.1 | 7.1 | 6.9 | 7.0 | 7.1 | 7.0 |
| Ammonia nitrogen | Less than 0.5 mg/L | ND [b] | ND | ND | ND | ND | ND |
| Total phosphorous | Less than 0.01 mg/L | ND | ND | ND | ND | ND | ND |
| Chlorine ion | Less than 250 mg/L | 18 | 19 | 22 | 22 | 19 | 21 |
| Total trihalomethane | Less than 0.1 mg/L | 0.018 | 0.016 | 0.023 | 0.031 | 0.034 | 0.036 |
| Residual chlorine | Less than 4.0 mg/L | 0.73 | 0.74 | 0.86 | 0.88 | 0.78 | 0.53 |
| Various bacteria | Less than 100 CFU/mL | 0 | 0 | 0 | 0 | 0 | 0 |
| Total coliform | No detection/100mL | ND | ND | ND | ND | ND | ND |
| Fluorine | Less than 1.5 mg/L | ND | ND | ND | ND | ND | ND |
| Aluminum | Less than 0.2 mg/L | 0.03 | 0.03 | ND | 0.02 | ND | 0.02 |
| **Inspection judgment** | | Suitability | | Suitability | | Suitability | |

[a] Nephelometric Turbidity Units; [b] Not Detected.

## 4. Conclusions

With the aim of replacing the existing mechanical rapid mixers in water treatment plants with non-powered hydraulic mixing, CFD was utilized to obtain the design factors (size, depth, hole size, and wing shape of the baffles) of non-powered hydraulic mixing. Then, the optimized non-powered hydraulic mixing with three layers of baffles and holes was developed and evaluated as an alternative to vertical shaft impellers in a rapid mixing process through both computational fluid dynamics (CFD) modeling and field applications. From the CFD modeling results, wider velocity distribution and stronger turbulence generated by non-powered hydraulic mixing successfully enhanced the mixing efficiency between coagulants and pollutants. Furthermore, the turbulence caused by excess kinetic energy between the inlet and second-layer baffles ensures rapid mixing of the coagulants throughout the total water flow and overcomes the damping effect of the components inside the mixing basin after the second-layer baffle. As the inlet velocity increased, the loss in both pressure and head between the inlet and outlet increased, indicating that the optimal inlet velocity needs to be determined for sufficient mixing without increases in both energy consumption and maintenance costs. Based on the appropriate design and analysis optimized by CFD modeling and the tradeoff between mixing efficiency and energy consumption, the optimal inlet velocity of 0.5 m/s was determined in this study to implement non-powered hydraulic mixing into the water treatment plants. Thus, CFD modeling can systematically optimize both velocity and pressure distribution and turbulence intensity and pathlines of hydraulic mixing in coagulation-flocculation processes using several critical design factors. Finally, no difference in water quality between a plant with non-powered hydraulic mixing and a plant with vertical shaft impellers from 2014 to 2016 using identical raw water and coagulant aluminum sulfate (VI) $[Al_2(SO_4)_3 \cdot 18H_2O]$ indicated that non-powered hydraulic mixing can be replaced with vertical shaft impellers, hence, both energy consumption and maintenance costs can be reduced. Consequently, the non-powered hydraulic mixing using horizontal jumps and flow energy developed in this study can replace the vertical shaft impellers in water treatment plants.

**Author Contributions:** Conceptualization, T.I.O. and J.S.C.; methodology, M.Y.Z.; software, M.Y.Z.; validation, J.C.J.; writing—original draft preparation, T.I.O. and M.Y.Z.; writing—review and editing, J.C.J.; visualization, M.Y.Z. All authors have read and agree to the published version of the manuscript.

**Funding:** This work was supported by the Korea Environment Industry and Technology Institute (KEITI) through Public Technology Program based on Environmental Policy Program (2016000710008) and Recycling Technology Program on Municipal Solid Waste(RE201906034) funded by the Korean Ministry of Environment.

**Conflicts of Interest:** The authors declare no conflict of interest.

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
