# Peer review of "Computational Fluid Dynamics Modeling and Field Applications of Non-Powered Hydraulic Mixing in Water Treatment Plants"

_water, doi:10.3390/w12040939_

Round 1

Reviewer 1 Report

Computational Fluid Dynamics Modeling and Field Applications of Non-Powered Hydraulic Mixing

Ohm et al.

A CFD model has been developed to simulate non-powered hydraulic mixing with three layers of baffles and holes. The model has been implemented on ANSYS-FLUENT. The inlet velocity has been varied and the effects of inlet velocity on the velocity distribution, internal pressure and turbulence intensity have been investigated. The authors have also found an optimum inlet velocity as a result of there is a trade-off between energy loss and mixing when the inlet velocity is increased. Authors have also presented a performance comparison between powered ad non-powered mixing by using field data and it indicates that non-powered mixing is advantageous for energy saving. The manuscript is well written and the work is very useful to understand energy optimization for water treatment plants. I suggest authors addressing following points before publishing this manuscript.

  1. Do authors have considered chemical reactions here? What is the purpose of the reaction term in Eq (1) and how is it calculated if there is one?
  2. It seems that the convergence criteria is not small enough compared to the inlet velocity. Will the results change significantly if the criteria is reduced by one order or so?
  3. Line 207-210: This conclusion drawn from Fig.4 is not convincing. Give more explanations with data or remove it.
  4. The main body text is not referred to Table 1. In Table 1, it would be better to normalize the last column, the difference between inlet and outlet velocity, by the inlet velocity and then make a conclusion. It is normal that velocity difference increases with inlet velocity due to head losses and therefore normalization would be important to draw a conclusion.
  5. Line 232-235: The statement is not convincing. Is this conclusion drawn from what results?
  6. It is not clear how the optimum velocity of 0.5m/s is calculated.

Author Response

Please see the attached file below for authors' response letter. 

Reviewer 2 Report

In the attachment I send suggestions and comments to the authors.

Best regards

Author Response

Please see the attached file below for authors' response letter.

Thank you so much. 

Round 2

Reviewer 2 Report

According to the reviewer the article can be published in its current form.